# Long term outcomes of patients with tuberculous meningitis: The impact of drug resistance

Emily E. Evans[1], Teona Avaliani[2], Mariam Gujabidze[2], Tinatin Bakuradze[2], Maia Kipiani[2], Shorena Sabanadze[2], Alison G. C. Smith[1], Zaza Avaliani[2], Jeffrey M. Collins[3], Russell R. Kempker[1,3]*

1 Emory University School of Medicine, Atlanta, Georgia, United States of America, 2 National Center for Tuberculosis and Lung Diseases, Tbilisi, Georgia, 3 Department of Medicine, Division of Infectious Diseases, Emory University, Atlanta, Georgia, United States of America

* rkempke@emory.edu

## Abstract

### Background

Little is known about the impact of drug-resistance on clinical outcomes among patients with tuberculosis meningitis (TBM).

### Methods

A retrospective cohort study among patients treated for TBM in Tbilisi, Georgia. We performed medical chart abstraction to collect patient data. Long-term vital status was assessed using the Georgia National Death Registry. We utilized a Cox proportional-hazards model to evaluate the association of drug-resistance and mortality.

### Results

Among 343 TBM suspects, 237 had a presentation consistent with TBM. Drug resistance was suspected (n = 5) or confirmed (n = 31) in 36 patients including 30 with multidrug- or rifampin-resistance and 6 with isoniazid-resistance. Thirty-four patients had HIV. The median follow-up time was 1331 days (IQR, 852–1767). Overall, 73 of 237 (30%) people died with 50 deaths occurring during and 23 after treatment. The proportion of death was higher among patients with drug-resistant vs. drug-susceptible disease (67% vs. 24%, p<0.001) and with HIV versus no HIV (59% vs 27%, p<0.001). Mortality was significantly higher in patients with drug-resistant TBM after 90 days of treatment (aHR = 7.2, $CI_{95\%}$ [3.6–14.3], p < 0.001).

### Conclusions

Mortality was high among patients with drug-resistant TBM with many deaths occurring post treatment. More effective treatment options are urgently needed for drug-resistant TBM.

**Data Availability Statement:** All relevant data are within the paper and its Supporting Information files.

**Funding:** This work was supported in part by IDSA and HIVMA through a G.E.R.M. award (E.E) as well as support from the NIH including the Fogarty International Center (D43 TW007124) and NIAID (R03 AI139871 [RRK], K23AI103044 [RRK], K23 AI144040[JMC], P30AI168386 [JMC, RRK]). The funders had no role in the study design, data collection and analysis, decision to publish or preparation of the manuscript.

**Competing interests:** The authors have declared that no competing interests exist.

# Introduction

Tuberculosis meningitis (TBM) is the most lethal manifestation of TB disease [1]. Among patients treated for drug susceptible TBM, overall hospital mortality rates are up to 50% and long-term five-year mortality rates of 58% [2,3]. Persons with TBM also suffer from high rates of long-term neurological sequelae due to disease complications including infarction, vasculitis, and hydrocephalus [4–6]. High rates of morbidity and mortality among patients with TBM are due in part to delays in diagnosis and hence prompt initiation of anti-TB treatment, as well as poor CSF penetration of many anti-TB drugs such as rifampin [7].

The emergence of drug–resistant (DR) TB including multidrug-resistant (MDR) and extensively drug-resistant (XDR) disease have presented major challenges to worldwide TB control [8]. Historically, clinical outcomes among patients with pulmonary MDR- or XDR-TB have been much worse compared to patients with drug-susceptible disease [9–11]. However, there are only limited data on the impact of drug-resistance on clinical outcomes among patients with TBM. To date there have been only a few studies of patients with drug-resistant TBM from a small number of geographic areas and none from Eastern European countries [12]. These limited data show mortality rates among patients treated for drug-resistant TBM ranges from 69–100% [5,13,14]. The advent of sensitive molecular diagnostics such as Xpert MTB/RIF, as well as implementation of new and repurposed drugs (i.e. bedaquiline, pretomanid, and linezolid) to treat MDR-TB present a tremendous opportunity to improve diagnosis and treatment outcomes in drug-resistant TBM. Thus, it is critical to establish a more accurate measure of current treatment outcomes in drug-resistant TBM against which future diagnostic and treatment interventions can be measured.

To address this question, we conducted a retrospective cohort study at a TBM referral hospital in the country of Georgia, which has a high burden of MDR-TB and a centralized system for collecting data on long-term mortality [15]. Our goal was to characterize long-term mortality in patients with TBM and determine the impact of drug resistance on morbidity and mortality.

# Methods

## Study setting

We utilized a retrospective observational cohort study design. Patients receiving treatment for TBM at the National Center for Tuberculosis and Lung Diseases (NCTLD) in Tbilisi, Georgia were included. The NCTLD campus contains the National Reference Laboratory, three inpatient hospitals and outpatient DOT clinics. There are twenty inpatient beds on the TBM ward and all patients with suspected TBM throughout the country are recommended to be referred to the NCTLD for initial evaluation and inpatient management. Patients admitted for TBM work up between January 1, 2013 and January 31, 2018 were eligible to be included. We excluded patients with an alternative diagnosis as determined by chart review from two U.S. based infectious diseases physicians not involved in patient care. Patients who did not meet the case definition of TBM by scoring criteria also had a detailed chart review and those without alternative diagnoses and a clinical presentation compatible with TBM were included. Standard treatment for drug-susceptible TBM during the study period consisted of 12 months of therapy including an intensive 2 month phase of rifampicin, isoniazid, pyrazinamide, an injectable agent (amikacin, kanamycin or streptomycin), and a fluoroquinolone (levofloxacin, moxifloxacin or ofloxacin) as well as dexamethasone and mannitol for management of intracranial hypertension. Drugs included in the continuation phase were determined by initial treatment response and included isoniazid, rifampin and ethambutol if there was a favorable

initial clinical and laboratory response and a more extensive regimen if otherwise. All inpatient care was provided by two dedicated TBM physicians, each with ≥ 30 years of experience. Treatment decisions for drug-resistant TBM were made by treating physicians and based on phenotypic or molecular drug-suscetibility testing (DST) results when available and/or clinical course. Patients were recommended to be hospitalized for initial TBM treatment until 1) they were no longer receiving an injectable agent and 2) they demonstrated substantial clinical improvement at which point care was transitioned to outpatient care in TB dispensaries. All treatment was provided by directly observed therapy (DOT). Regarding treatment for human immunodeficiency virus (HIV), patients receiving antiretroviral therapy (ART) at the time of TBM diagnsosis were continued on ART, while those not receiving ART were recommended to start ART 2–8 weeks after TBM treatment commenced. HIV treatment decisions were guided by consultant HIV physicians from the Georgia Infectious Diseases, AIDS, and Clinical Immunology Research Center (IDACIRC).

## Laboratory

All patients had a diagnostic lumbar puncture performed to obtain cerebrospinal fluid (CSF) for microbiological and molecular testing at the time of admission to the NCTLD. Microbiologic testing of CSF included Ziehl-Neelsen microscopy and mycobacterial culture on both Löwenstein-Jensen (LJ) solid and Mycobacterial Growth Indicator Tube (MGIT) liquid media. In cases with a positive culture for *Mycobacterium tuberculosis*, isolates underwent testing with the MTBDR*plus* line probe assay (Hain Lifescience, Nehren, Germany) and phenotypic DST as previously described [16]. Starting in April 2015, testing with the Xpert MTB/RIF assay (Cepheid, Sunnyvale, CA, USA) was implemented for all baseline CSF samples. Approximately 1–4 ml of non-centrifuged CSF was used and the test was performed according to standard procedures. General testing including CSF bacterial cultures, complete blood count and chemistry evaluations were carried out per Georgia NTP guidelines.

## Data collection

Patient medical information was abstracted from inpatient medical charts, the National Reference Laboratory database and the NCTLD electronic database. Additional HIV treatment information was obtained from the Infectious Diseases, AIDS, and Clinical Immuniology Research Center, (IDACIRC) which oversees the National HIV program. All patients treated for TBM without a proven or probable alternative etiology were included in the study, and patients were categorized as either 1) not TBM, 2) possible TBM, 3) probable TBM or 4) definite TBM according to the uniform TBM research case definition. In brief, this case definition uses clinical, CSF, and imaging criteria to provide a score that categorizes patients into possible, probable, or definite TBM [17]. TBM grade was defined as 1, normal mental status with no focal neurological deficits; 2, either normal mental status with focal neurological deficits or mildly altered mental status and 3, severe altered mental status. Patients were categorized as either drug susceptible or drug-resistant TBM with drug-resistance defined as having INH monoresistance (without rifampin), rifampin mono-resistance, MDR, pre-XDR or XDRbased on NCTLD and WHO guidance from 2020. MDR was defined as resistance to rifampin and isoniazid; pre-XDR as additional resistance to a fluoroquinolone or injectable second-line agent; XDR as resistance to both a flouroquinolone and an injectable second-line agent[18]. Drug resistance was defined End of treatment or initial outcomes were defined as completed, not evaluated, lost to follow-up (LTFU), failure, and death. An outcome of not evaluated occurred when patient care was transferred outside of the NCTLD, and the final outcome was unknown. LTFU was defined as a treatment interruption of ≥2 months. Data on long term

mortality including date of death if applicable was obtained from the Georgia National Office of Statistics with the last date of checking being August 29, 2019. Data collection forms were developed in accordance with standardized methods put forth by the Tuberculosis International Research Consortium [19]. All data was collected on paper forms and then entered into an online REDCap database [20]. This study was approved by the Institutional Review Board both at the NCTLD and Emory University.

### Statistical analysis

Statistical analyses were performed using SAS, version 9.4 (SAS Institute, Cary NC), and R version 3.6.0. For comparisons of baseline characteristics among patients treated for drug-susceptible versus drug-resistant disease, a chi-square test was used for categorical variables and a two-sided independent T-test was used for continuous variables. Kaplan-Meier curves were generated for the entire cohort and among patients categorized as "probable" and "definite" TBM to compare time to death among patients treated for drug-resistant and drug-susceptible TBM.

We calculated survival for drug-resistant and drug susceptible TBM according to the Kaplan–Meier method and used a Cox proportional-hazards model to adjust for between-group differences in baseline characteristics [21,22].We created a model using both the entire cohort and a model limited to cases identified as either "definite" or "probable" TBM. The main predictor variable was treatment for drug-resistant disease. Other factors with clinical significance in univariate analysis, with biological plausibility, or previously found to be associated with mortality among patients with TBM were considered in the model. The initial Cox model violated the proportional hazards assumption due to a differential effect of drug-resistant TBM on short-term mortality (≤ 90 days after treatment start) and long-term mortality (>90 days after treatment start). Thus, the hazard ratio of drug-resistant TBM versus drug-susceptible TBM was calculated for both short-term and long-term mortality. A p-value <0.05 was considered statistically significant.

### Results

Among 343 admissions for suspected TBM, 237 persons met eligibility criteria for study inclusion. The most common reasons for exclusion were pulmonary TB without clear evidence of central nervous system (CNS) involvement and the presence of an alternative diagnosis (Fig 1). There were 90 patients (38%) with either a CNS or non-CNS sample positive for *M. tuberculosis* by Xpert TB/RIF or culture testing. Among the 47 patients (20%) with a positive CNS sample for *M.tb*, 29 (12%) had a positive CSF Xpert MTB/RIF, 38 (16%) had a positive CSF culture and 1 (0.4%) was diagnosed by brain biopsy. Forty-eight (20%) TBM cases were classified as definite TBM, 31 (13%) as probable TBM, 153 (65%) as possible TBM and 5 (2%) as unlikely TBM. There were 36 TBM cases treated for drug-resistant disease including 31 with confirmed resistance (Fig 1). Among confirmed drug-resistant cases, 6 were INH resistant, 2 were rifampicin mono-resistant, 7 were MDR, 9 were pre-XDR and 7 were XDR. The five unconfirmed cases were treated for MDR TBM given either prior TB treatment failure or a known contact with confirmed MDR TB.

Among the 237 patients with TBM, 143 (60%) were men and the mean age was 44 years. There were 34 patients (14%) co-infected with HIV with a median (IQR) CD4+ T-cell count of 40 (20, 110) cells/ul. For most patients (83%), this was the first episode of TB. Notably, patients treated for drug-resistant versus drug-susceptible TBM were more likely to have HIV (50% vs. 8%, p < .01) and to have had prior TB (56% vs. 10%, p < .01). The most common presenting manifestations were headache, nuchal rigidity, fever, and altered mental status. The median

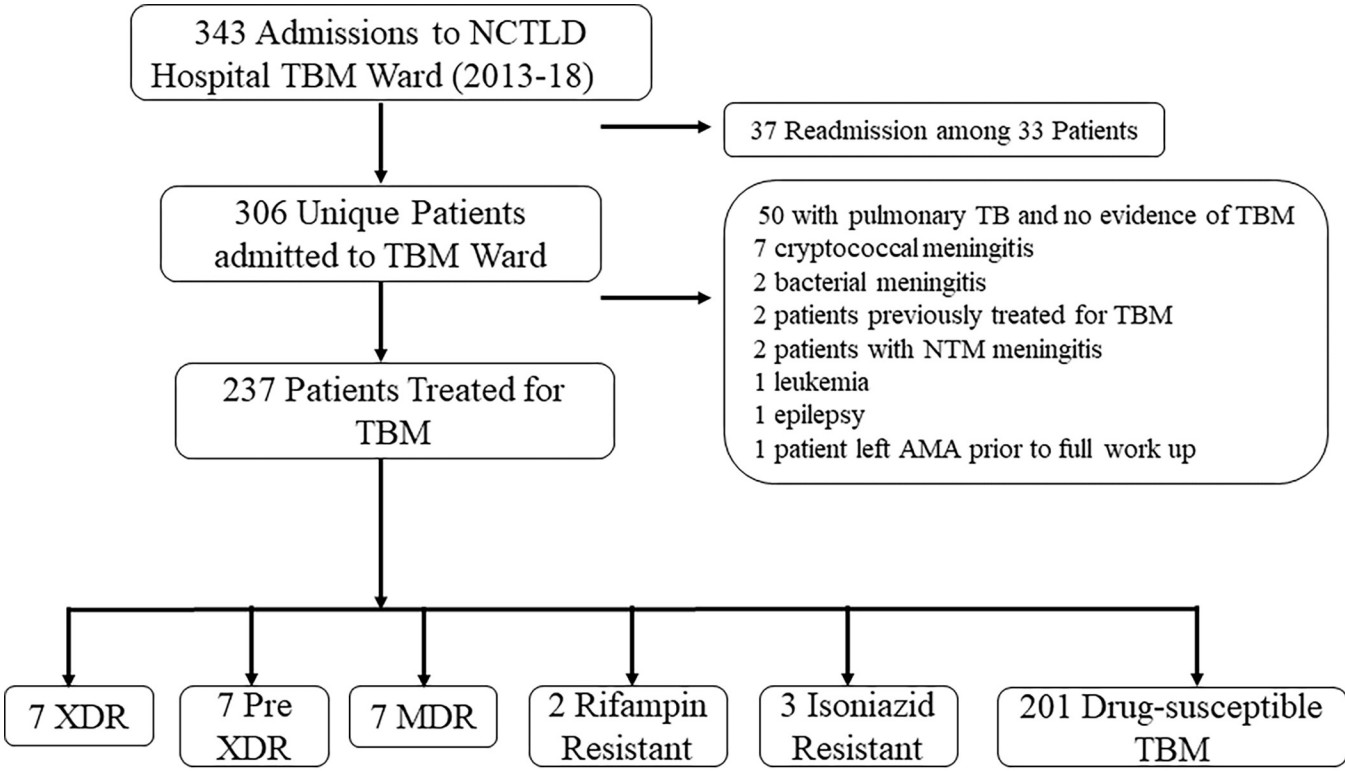

**Fig 1. Flowchart of eligible and included patients treated for tuberculosis meningitis by drug-resistance pattern.**

CSF WBC at baseline was 213 cell/μl. For baseline characteristics of the cohort and by treatment category see Table 1.

## Treatment

The anti-TB drugs used throughout treatment are detailed in Table 2. Patients with drug resistant-disease were more likely to receive moxifloxacin, cycloserine, para-aminosalicyclic acid, and capreomycin. There was little use of newly implemented drugs (linezolid, clofazimine, imipenem, bedaquiline, and delaminid) among patients with drug-resistant disease especially as part of initial treatment. Regarding HV treatment, 14 of 34 (41%) of HIV-infected persons were receiving ART at the time of TBM diagnosis and all were continued on treatment; four additional HIV-infected persons started ART during TBM.

## Clinical outcomes

There were 73 deaths (31%) among our entire TBM cohort including 50 persons who died during treatment and 23 who died after treatment completion. There were 15 persons who were LTFU during treatment after a median of 62 days (IQR 39–138), 12 of whom were determined to have died by death records. There was a high death rate among people with drug-resistant TBM (67%), HIV (59%) and Grade 3 TBM disease (58%). Among the 31 cases of confirmed drug-resistant TBM, the death rate was 77% including all 7 patients with XDR disease.

**Table 1. Characteristics and outcomes of patients with tuberculosis meningitis by treatment type characteristics.**

|  | Overall n = 237 (%) | Drug Susceptible n = 201 (%) | Drug Resistant n = 36 (%) | p-value* |
|---|---|---|---|---|
| Mean age, years (SD) | 44 (17) | 45 (18) | 40 (9) | 0.28 |
| Male sex | 143 (60) | 117 (58) | 26 (72) | 0.16 |
| Weight < 50 kilograms | 12 (4) | 11 (5) | 1 (3) | >0.99 |
| History of imprisonment | 24 (10) | 15 (7) | 9 (25) | <0.01 |
| Contact to active tuberculosis case | 25 (11) | 22 (11) | 3 (8) | 0.86 |
| **HIV** | | | | |
| HIV infected | 34 (14) | 16 (8) | 18 (50) | <0.01 |
| New diagnosis | 12 (5) | 10 (5) | 2 (6) | >0.99 |
| CD4, cells/ul, median (IQR) [n = 34] | 40 (20, 110) | 63 (30, 173) | 31 (18, 54) | 0.11 |
| Currently on ART | 14 (6) | 6 (3) | 8 (26) | <0.01 |
| **Co-morbidities** | | | | |
| Hepatitis C virus antibody positive | 31 (13) | 14 (7) | 17 (47) | <0.01 |
| History of intravenous drug use | 17 (7) | 6 (3) | 11 (31) | <0.01 |
| Alcohol Use Disorder | 9 (4) | 6 (3) | 3 (8) | 0.14 |
| Diabetes | 12 (5) | 10 (5) | 2 (6) | >0.99 |
| **Case Definition** | | | | |
| New Case | 197 (83) | 181 (90) | 16 (44) | <0.01 |
| Relapse | 11 (5) | 9 (4) | 2 (6) | |
| Treatment after LTFU | 12 (5) | 3 (1) | 9 (25) | |
| Treatment after failure | 5 (2) | 1 (1) | 4 (11) | |
| Other | 12 (5) | 7 (3) | 5 (14) | |
| **Presentation** | | | | |
| Neurological Status | | | | 0.01 |
| TBM Grade 1/Normal | 33 (14) | 32 (16) | 1 (3) | |
| TBM grade 2/Mild | 142 (60) | 123 (62) | 19 (54) | |
| TBM grade 3/Severe | 60 (26) | 45 (23) | 15 (43) | |
| Unknown | 2 | 1 | 1 | |
| Days since first neurologic symptom, median (IQR) | 10 (7, 15) | 10 (7, 15) | 10 (7, 14) | 0.95 |
| Altered Mental Status | 195 (82) | 163 (81) | 32 (89) | |
| Fever | 214 (90) | 183 (91) | 31 (86) | 0.42 |
| Headache | 217 (92) | 186 (93) | 31 (86) | 0.29 |
| Vomiting | 147 (62) | 132 (66) | 15 (42) | 0.02 |
| Nuchal rigidity | 219 (92) | 188 (94) | 31 (86) | 0.14 |
| Seizures | 23 (10) | 19 (10) | 4 (11) | 0.39 |
| Cranial nerve palsy | 63 (10) | 57 (28) | 6 (17) | 0.23 |
| Urinary retention | 51 (22) | 43 (21) | 8 (22) | 0.62 |
| Hemiplegia | 17 (7) | 12 (6) | 5 (14) | 0.22 |
| Paraplegia | 18 (8) | 16 (8) | 2 (6) | >0.99 |
| **Site of Disease** | | | | |
| Evidence of extra-CNS disease | 70 (30) | 44 (22) | 26 (72) | <0.01 |
| Pulmonary disease | 60 (25) | 35 (15) | 25 (69) | <0.01 |
| **Care prior to admission** | | | | |
| Received antibiotic therapy prior to admission | 88 (41) | 79 (41) | 9 (35) | 0.15 |
| **Baseline CSF data** | | | | |
| CSF WCC, cells/ul, mean (SD) | 213 (263) | 226 (280) | 144 (116) | 0.09 |
| CSF protein, mg/dl, mean (SD) | 177 (294) | 175 (305) | 189 (229) | 0.15 |
| CSF glucose, mg/dl, mean (SD) | 42 (24) | 44 (24) | 31 (20) | <0.01 |

*(Continued)*

**Table 1.** (Continued)

| | Overall n = 237 (%) | Drug Susceptible n = 201 (%) | Drug Resistant n = 36 (%) | p-value* |
|---|---|---|---|---|
| **Microbiology** | | | | |
| Microbiologic confirmation at any site | 90 (38) | 59 (29) | 31 (86) | <0.01 |
| Microbiologic confirmation in CNS sample | 48 (20) | 26 (13) | 22 (61) | <0.01 |
| CSF Xpert positive | 29 (12) | 14 (7) | 15 (42) | <0.01 |
| CSF culture positive | 38 (16) | 19 (10) | 19 (53) | <0.01 |
| Microbiologic confirmation from non-CNS site only | 42 (18) | 33 (16) | 9 (25) | 0.31 |
| Clinical diagnosis | 147 (62) | 142 (71) | 5 (14) | <0.01 |
| **Imaging** | | | | |
| Either MRI brain or CT head performed | 166 (70) | 144 (72) | 22 (61) | 0.28 |
| Any CNS imaging abnormality | 135 (57) | 114 (57) | 21 (58) | >0.99 |
| **Mean baseline laboratory data** | | | | |
| Hemoglobin (SD) | 12.3 (2.0) | 12.4 (1.9) | 11.5 (2.2) | 0.02 |
| Sodium (SD) | 131 (8) | 132 (8) | 129 (8) | 0.14 |
| Creatinine (SD) | 79 (26) | 81 (27) | 67 (15) | <0.01 |
| Albumin (SD) | 34 (7) | 34 (7) | 34 (6) | 0.53 |
| **Case definition[1]** | | | | <0.01 |
| Definite TBM | 48 (20) | 26 (13) | 22 (61) | |
| Probable | 31 (13) | 24 (12) | 7 (19) | |
| Possible | 153 (65) | 146 (73) | 7 (19) | |
| Unlikely | 5 (2) | 5 (2) | 0 | |
| **Initial treatment outcome** | | | | |
| Cure/Completed | 148 (63) | 140 (69) | 8 (22) | <0.01 |
| Lost to follow up | 15 (6) | 13 (6) | 2 (6) | |
| Treatment failure | 4 (2) | 4 (2) | 0 (0) | |
| Death | 50 (21) | 29 (14) | 21 (58) | |
| Unknown or not evaluated | 20 (8) | 15 (7) | 5 (14) | |
| **Long term mortality** | | | | |
| Death | 73 (31) | 49 (24) | 24 (67) | <0.01 |
| Alive | 161 (68) | 150 (75) | 11 (31) | |
| Not determined | 3 (1) | 2 (1) | 1 (3) | |
| Died after treatment completion | 23 (10) | 20 (10) | 3 (8) | |
| **Follow-up time, days** (Median, IQR) | 1331 (852–1767) | 1390 (952–1779) | 503 (70–1642) | 0.01 |

* **Statistical tests performed:** Wilcoxon rank-sum test; chi-square test of independence; Fisher's exact test.

**Abbreviations used:** SD, standard deviation; HIV, human immunodeficiency virus; ART, anti-retroviral therapy; TBM, tuberculous meningitis; CNS, central nervous system; MRI, magnetic resonance imaging; CT, computed tomography; CXR, chest X-ray; LTFU, lost to follow up; ATT, anti-tuberculosis therapy; CSF, cerebrospinal fluid; WCC, white cell count; *Mtb*, *Mycobacterium tuberculosis*.

1. Defined by the Uniform Tuberculous Meningitis Research Case Definition Criteria.

Among persons with HIV, the death rate among those on ART at TBM diagnosis was 64% and 55% for those not receiving ART. **S1 Table** compares patient characteristics by vital status.

## Drug resistance and mortality

The proportion of persons who died was significantly higher in persons treated for drug-resistant TBM versus those with drug-susceptible TBM (67% vs. 24%, p<0.001; **Fig 2A** and **2B**). The difference in mortality was similar when we limited the analysis to persons with probable

**Table 2. Drugs included in baseline treatment regimens and ever in treatment, by drug resistance status.**

| Drug | Baseline regimen, all patients (N = 237) | Ever in regimen, all patients | Initial regimen, DS (N = 201) | Ever in regimen, DS | Initial regimen, DR (N = 36) | Ever in regimen, DR |
|---|---|---|---|---|---|---|
| **Any fluoroquinolone** | 220 (92.8) | 229 (96.6) | 190 (94.5) | 196 (97.5) | 30 (83.3) | 33 (91.7) |
| **Any injectable agent** | 216 (91.1) | 221 (93.2) | 192 (95.5) | 194 (96.5) | 24 (66.7) | 27 (75) |
| Isoniazid | 207 (87.3) | 212 (89.5) | 191 (95) | 196 (97.5) | 16 (44.4) | 16 (44.4) |
| Rifampin | 199 (84) | 205 (86.5) | 182 (90.5) | 188 (93.5) | 17 (47.2) | 17 (47.2) |
| Amikacin | 194 (81.9) | 202 (85.2) | 172 (85.6) | 177 (88.1) | 22 (61.1) | 25 (69.4) |
| Ofloxacin | 190 (80.2) | 197 (83.1) | 172 (85.6) | 176 (87.6) | 18 (50) | 21 (58.3) |
| Ethambutol | 138 (58.2) | 198 (83.5) | 120 (59.7) | 178 (88.6) | 18 (50) | 20 (55.6) |
| Pyrazinamide | 138 (58.2) | 156 (65.8) | 118 (58.7) | 133 (66.2) | 20 (55.6) | 23 (63.9) |
| Levofloxacin | 50 (21.1) | 180 (75.9) | 40 (19.9) | 158 (78.6) | 10 (27.8) | 22 (61.1) |
| Kanamycin | 39 (16.5) | 165 (69.6) | 36 (17.9) | 155 (77.1) | 3 (8.3) | 10 (27.8) |
| Moxifloxacin | 26 (11) | 39 (16.5) | 16 (8) | 20 (10) | 10 (27.8) | 19 (52.8) |
| Cycloserine | 25 (10.5) | 41 (17.3) | 5 (2.5) | 10 (5) | 20 (55.6) | 31 (86.1) |
| Para-aminosalicylic acid | 17 (7.2) | 31 (13.1) | 1 (0.5) | 2 (1) | 16 (44.4) | 29 (80.6 |
| Capreomycin | 15 (6.3) | 27 (11.4) | 0 (0) | 0 (0) | 15 (41.7) | 27 (75) |
| Prothionamide | 14 (5.9) | 22 (9.3) | 0 (0) | 0 (0) | 14 (38.9) | 22 (61.1) |
| Linezolid | 5 (2.1) | 10 (4.2) | 1 (0.5) | 1 (0.5) | 4 (11.1) | 9 (25) |
| Clofazimine | 4 (1.7) | 10 (4.2) | 0 (0) | 0 (0) | 4 (11.1) | 10 (27.8) |
| Streptomycin | 4 (1.7) | 14 (5.9) | 1 (0.5) | 9 (4.5) | 3 (8.3) | 5 (13.9) |
| Imipenem | 3 (1.3) | 7 (3) | 2 (1) | 2 (1) | 1 (2.8) | 5 (13.9) |
| Bedaquiline | 0 (0) | 3 (1.3) | 0 (0) | 0 (0) | 0 (0) | 3 (8.3) |
| Delamanid | 0 (0) | 3 (1.3) | 0 (0) | 0 (0) | 0 (0) | 3 (8.3) |
| **Hydrocephalus management** | | | | | | |
| | **In regimen, all patients** | | **In regimen, DS (N = 201)** | | **In regimen, DR (N = 36)** | |
| Mannitol | 226 (95.4) | | 191 (95) | | 35 (97.2) | |
| Furosemide | 35 (14.8) | | 29 (14.4) | | 6 (16.7) | |
| Ventriculoperitoneal shunt | 2 (0.8) | | 2 (1) | | 0 (0) | |
| **Anti-inflammatory therapies** | | | | | | |
| Dexamethasone | 232 (97.9) | | 197 (98) | | 35 (97.2) | |
| Prednisone | 35 (14.8) | | 29 (14.4) | | 6 (16.7) | |

* Baseline was defined as receiving within 14 days of admission.

Abbreviations: DS, drug susceptible; DR, drug-resistant.

or definite TBM (74% vs 26%, p<0.001; **Fig 2C** and **2D**) and when limited to those with microbiologically confirmed TBM (77% vs. 19%, p<0.001; **S1 Fig**). In our multivariate model, there was a non-significant trend towards increased mortality in persons treated for drug-resistant TBM versus drug-susceptible TBM in the first 90 days of treatment (aHR 1.57, 95% CI 0.65–3.77), while beyond 90 days treatment for drug-resistant TBM was associated with a higher rate of death (aHR 7.15, 95% CI 3.58–14.31; **Table 3**).When limited to "definite" or "probable" TBM cases, we similarly observed a non-significant increase in mortality in persons with drug-resistant TBM before 90 days (aHR 1.24, 95% CI 0.42–3.65) and a significant increase in mortality after 90 days (aHR 9.2, 95% CI 2.86–29.63; **S2 Table**).Other significant risk factors for mortality in multivariate analysis included being HIV positive (aHR 2.89, 95% CI 1.52–5.49), age (aHR 1.06 per year, 95% CI 1.04–1.08), and Grade 3 TBM (aHR 2.26, 95% CI 1.39–3.68). When we limited cases to only definite and probable TBM, HIV status and

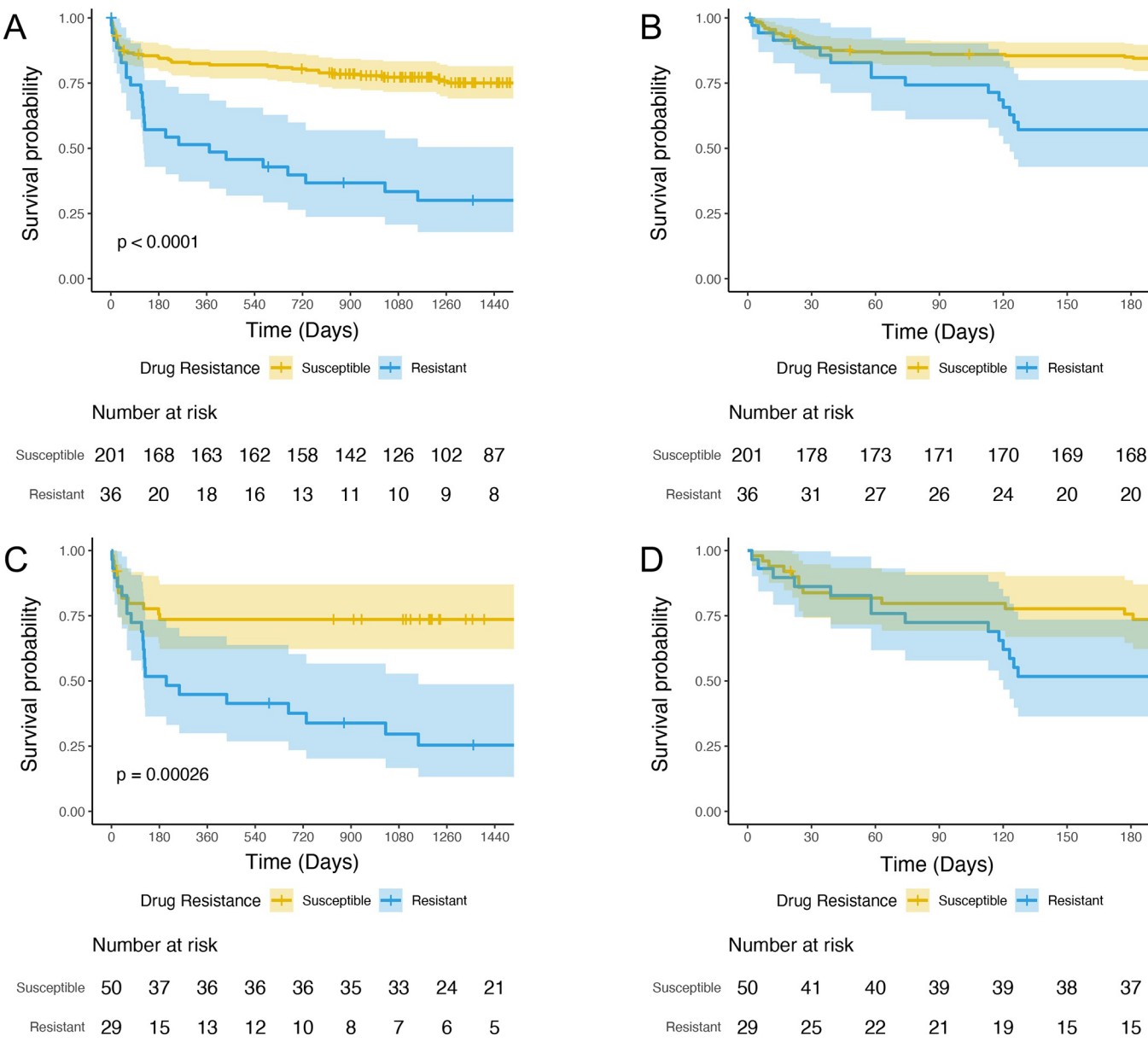

**Fig 2.** Kaplan Meier survival curves for (A) all patients treated for tuberculosis meningitis (TBM; n = 237) over the full period of observation and (B) over first 6 months. (C) Probable and definite TBM cases (n = 79) over the full period of observation and (D) over first 6 months of treatment. Mortality in persons treated for drug susceptible-TBM is depicted by the yellow line and for persons with drug resistant-TBM by the blue line. The shaded region surrounding the line depicts the 95% confidence interval of the mortality estimate.

TBM Grade of 3 were no longer statistically significant (aHR 2.13, 95% CI 0.98–4.67 and aHR 2.05, 95% CI 0.97–4.34, respectively).

## Discussion

In one of the largest cohorts of patients with drug-resistant TBM published to date, we found drug-resistant TBM was associated with a significantly higher risk of death versus drug-susceptible TBM, with a long-term mortality of 67%. We further found that this significant increase in mortality among patients with drug-resistant TBM occurred >90 days after treatment

**Table 3. Multivariate analysis of predictors of mortality among patients treated for tuberculosis meningitis.**

| Variables | Adjusted Hazards Ratio | 95% CI | p-value |
|---|---|---|---|
| Treated for drug-resistant TBM | | | |
| < 90 days | 1.57 | 0.65, 3.77 | 0.32 |
| > 90 days | 7.15 | 3.58, 14.31 | <0.001 |
| HIV | 2.89 | 1.52, 5.49 | 0.001 |
| Male sex | 0.94 | 0.56, 1.57 | 0.81 |
| Age, per year | 1.06 | 1.04, 1.08 | <0.001 |
| Grade 3 TBM* | 2.26 | 1.39, 3.68 | 0.001 |

Abbreviations: HIV, human immunodeficiency virus; TBM, tuberculous meningitis

*Grade 1 and 2 TBM were analyzed together.

initiation, with an observed > 7-fold higher risk of death. Additionally, we discovered that >30% of all deaths in TBM patients occurred after treatment completion and that 80% of patients LTFU subsequently died. These findings highlight the marked increase in mortality associated with drug-resistant TBM in patients receiving currently available treatment regimens and simultaneously the urgent to need to develop new treatment regimens. Moreover, our findings demonstrate the critical need for long-term follow up (>1 year) among patients with TBM to determine post treatment death rates and identify areas for possible intervention.

As indicated by our clinical outcomes and those from external reports, the treatment of TBM is suboptimal. A recent meta-analyses of >6,000 patients, found an overall risk of death of 25% at 12 months among patients treated for TBM [23]. Furthermore, the authors found no substantial improvement in outcomes over time, a finding supported by our death rate of 24% during treatment among patients treated for drug-susceptible TBM. The lack of improvement in clinical outcomes is not surprising given the limited progress in developing new treatments for TBM. Treatment regimens for TBM were designed primarily by extrapolating pulmonary TB treatment recommendations; however, this approach has drawbacks [1]. Importantly, two of the first-line anti-TB drugs, rifampicin and ethambutol, have poor penetration into the CSF, thus limiting their potential effectiveness [7]. Due to these pharmacokinetic limitations, recent studies have focused on adding a fluoroquinolone given their high penetration into the CSF. Unfortunately, two recent randomized clinical trials failed to show a benefit when adding a fluoroquinolone to standard first-line drugs [13,24]. In line with these findings, most of our cohort received an intensified regimen, including an injectable agent and a fluoroquinolone; however, our mortality rates remained high. Another key difference in pulmonary TB and TBM, is the role of the host inflammatory response in determining outcomes. Eloquent studies have demonstrated an important role of the eicosanoid inflammatory pathway (including both hyper and hypoinflammatory responses) and tryptophan metabolism in determining outcomes including neurological disability [25,26]. The role of the host inflammatory response in determining outcomes also provides a potential rationale why the impact of drug-resistant TBM on outcomes was not seen until >90 days. Similarly, another study found isoniazid resistance was not associated with death until after 60 days of treatment among patients with TBM [5]. In summary, our results indicate that even when employing rapid diagnostic testing methods (Xpert), intensified treatment regimens and recommended anti-inflammatory agents that TBM mortality rates remain unacceptably high and improved treatments are needed.

A striking finding from our study is the extremely high death rate among TBM patients with HIV (59%) and/or those treated for drug-resistant TBM (67%). Our high rate of death among TBM patients with HIV was similar to the mortality rate among other HIV positive

cohorts (48–67%) as reported in the above mentioned meta-analysis, and calls for further efforts at prevention and treatment among this high risk group [23]. Our alarming death rate among patients treated for drug-resistant disease is similar to the few available reports and demonstrates the futility of currently available treatments. Among two separate studies from Vietnam, one found all 10 patients (100%) with MDR-TBM disease died, while the other found 11 of 16 (69%) died [13,14]. In the latter study, a higher dose of rifampin along with levofloxacin was added to standard first-line therapy. Among a long-term follow study of TBM patients in New York City treated from 1992–2001, almost all of the 67 patients with rifampin-resistant or MDR disease died (94%) [5]. Our study provides important modern outcome data; specifically, high rates of poor outcomes despite approximately half of drug-resistant TBM patients diagnosed with a rapid molecular diagnostic test (Xpert MTB/RIF) and most receiving aggressive and intensified drug regimens. Our report also documents the first clinical outcomes of patients with confirmed XDR-TBM and that an intensified treatment regimens had no benefit. Notably, the study period was prior to the programmatic implementation of bedaquiline, delamanid, linezolid and imipenem into treatment regimens for TBM at the NCTLD and thus few patients received.

Among recently implemented anti-TB drugs, linezolid is an attractive option to treat TBM given its high penetration into the CSF and preliminary clinical data finding short term use is associated with improved clinical outcomes [27,28]. Currently, there are two prospective clinical trials assessing the use of linezolid for TBM (NCT03927313, NCT04021121) which will provide much need data to guide use. While bedaquiline has led to a substantial improvements in drug-resistant pulmonary TB outcomes, its utility for TBM is uncertain as a high protein binding may limit its ability to cross the blood brain barrier as demonstrated by a report of unmeasurable CSF concentrations from a patient with TBM [29]. In contrast, data from 6 rabbits and 3 humans CSF on delamanid found free drug concentrations are generally higher in the CSF than plasma and higher than critical concentrations [30]. In addition to the urgent need for clinical trials and pharmacokinetic studies of these recently implemented and novel drugs further investigations into better understanding the host inflammatory response to TBM are needed to guide development of host directed therapies.

Our study was limited by a low rate of microbiologically confirmed TBM cases (38%); however, the majority of patients (86%) with drug-resistant TBM had microbiological confirmation of disease. Given the low sensitivity of current methods utilized to TBM this is a limitation common to TBM studies. When limiting analyses to patients with microbiologically confirmed disease, our findings were similar which strengthen the validity of our main result that patient with drug-resistant TBM experience worse outcomes. As centrifugation of higher CSF volumes has been shown to improve the yield CSF Xpert texting [31], our use of low volume non centrifuged samples may have resulted in missed cases. Given the retrospective nature of our study, we may have not accounted for potential confounders. Lastly, lack of information regarding detailed HIV treatment precluded the ascertainment of impact of effective ART on clinical outcomes, immune reconstitution inflammatory syndrome and other ART-associated complications.

In summary, we observed an overall high mortality among patients with TBM and a significantly higher long-term mortality in patients with drug-resistant TBM compared to those with DS-TBM, even when adjusted for well known risk factors. Our findings demonstrate that early diagnosis of drug-resistance and intensified regimens are not enough and consequently the urgent need for improved treatment regimens for TBM, particularly drug-resistant TBM.

## Supporting information

**S1 Fig.** Kaplan Meier survival curves for (A) all patients with microbiologically confirmed tuberculosis meningitis (TBM; n = 48) over the full period of observation and (B) over first 6

months. Mortality in persons with drug susceptible-TBM is depicted by the yellow line and in persons with drug resistant-TBM by the blue line. The shaded region surrounding the line depicts the 95% confidence interval of the mortality estimate.
(TIF)

**S1 Table. Characteristics of patients with tuberculosis meningitis by vital status.** Abbreviations used: HIV, human immunodeficiency virus; ART, anti-retroviral therapy; TB, tuberculosis; TBM, tuberculous meningitis; CNS, central nervous system; MRI, magnetic resonance imaging; CT, computed tomography; CXR, chest X-ray; LTFU, lost to follow up; ATT, anti-tuberculosis therapy; CSF, cerebrospinal fluid; WCC, white cell count. Note: Three patients excluded for whom final outcome was not determined. 1. Defined by the Uniform Tuberculous Meningitis Research Case Definition Criteria.
(DOCX)

**S2 Table. Multivariate analysis of predictors of mortality for patients with probable or definite tuberculosis meningitis.** Abbreviations: DR, drug resistant; HIV, human immunodeficiency virus; TBM, tuberculous meningitis.
(DOCX)

## Acknowledgments

We are thankful for the contributions of Nino Ruxadze at the HIV center in Tbilisi, Georgia for their collaboration to obtain HIV specific data.

## Author Contributions

**Conceptualization:** Emily E. Evans, Maia Kipiani, Shorena Sabanadze, Russell R. Kempker.

**Data curation:** Emily E. Evans, Teona Avaliani, Mariam Gujabidze, Tinatin Bakuradze, Maia Kipiani.

**Formal analysis:** Emily E. Evans, Alison G. C. Smith, Jeffrey M. Collins.

**Investigation:** Emily E. Evans, Teona Avaliani, Mariam Gujabidze, Tinatin Bakuradze, Maia Kipiani, Shorena Sabanadze, Zaza Avaliani, Russell R. Kempker.

**Methodology:** Emily E. Evans, Tinatin Bakuradze, Maia Kipiani, Shorena Sabanadze, Jeffrey M. Collins.

**Project administration:** Emily E. Evans, Teona Avaliani, Mariam Gujabidze, Maia Kipiani, Zaza Avaliani.

**Resources:** Zaza Avaliani.

**Supervision:** Tinatin Bakuradze, Shorena Sabanadze, Zaza Avaliani.

**Writing – original draft:** Emily E. Evans, Russell R. Kempker.

**Writing – review & editing:** Teona Avaliani, Mariam Gujabidze, Tinatin Bakuradze, Maia Kipiani, Shorena Sabanadze, Alison G. C. Smith, Zaza Avaliani, Jeffrey M. Collins, Russell R. Kempker.

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
