## [Decision Letter · Decision Letter 0]

7 Mar 2022

PONE-D-21-30903Long Term Outcomes of Patients with Tuberculous Meningitis: The Impact of Drug ResistancePLOS ONE

Dear Dr. Kempker,

Thank you for submitting your manuscript to PLOS ONE. After careful consideration, we feel that it has merit but does not fully meet PLOS ONE’s publication criteria as it currently stands. Therefore, we invite you to submit a revised version of the manuscript that addresses the points raised during the review process.

 There is still some statistical work to be done, together with some explanations about  unexplained data.

We look forward to receiving your revised manuscript.

Kind regards,

Pere-Joan Cardona, MD, PhD

Academic Editor

PLOS ONE

“This work was supported in part by IDSA and HIVMA through a G.E.R.M. award (E.E) as well as support from the NIH including the Fogarty International Center (D43 TW007124) and NIAID (R03 AI139871, K23AI103044, and K23 AI144040).”

“This work was supported in part by IDSA and HIVMA through a G.E.R.M. award (E.E) as well as support from the NIH including the Fogarty International Center (D43 TW007124) and NIAID (R03 AI139871 [RRK], K23AI103044 [RRK], and K23 AI144040[JMC). The funders had no role in the study design, data collection and analysis, decision to publish or preparation of the manuscript.”

Reviewers' comments:

Reviewer's Responses to Questions

**Comments to the Author**

1. Is the manuscript technically sound, and do the data support the conclusions?

Reviewer #1: Yes

Reviewer #2: Partly

2. Has the statistical analysis been performed appropriately and rigorously? 

Reviewer #1: Yes

Reviewer #2: I Don't Know

3. Have the authors made all data underlying the findings in their manuscript fully available?

Reviewer #1: Yes

Reviewer #2: Yes

4. Is the manuscript presented in an intelligible fashion and written in standard English?

Reviewer #1: Yes

Reviewer #2: Yes

5. Review Comments to the Author

Reviewer #1: Overall the manuscript has been well written.

In the results the authors have mentioned drug resistant TB is more common among HIV patients and those treated prior for TB. Can you please perform a statistical test to confirm the validity of these findings and mention the p value?

Thank you,

Reviewer #2: Long Term Outcomes of Patients with Tuberculous Meningitis: The Impact of Drug

Resistance.

Evans et al present an interesting retrospective study on the outcome of DR-TBM. The paper is well written, easy to understand for readers and (although many things have changed in DR-TB after ending the follow-up period, 2018) results obtained are interesting in order to better understand outcome of DR-TBM.

I have some suggestions that may improve de quality of the paper:

- Basal characteristics of patients with drug-resistant TB seem to be of worse prognosis than patients with drug-susceptible TB (higher proportion of HIV coinfection, worse neurologic basal status, evidence of extra-CNS disease…) so that may confound for interpreting the outcome. I would suggest the authors to explain in more detail how multivariate analysis was performed, variables included in the multivariate model, and including in table-1 the “p value” on differences of basal characteristics.

- It would be of interesting to include also a few data of the outcome of INH-resistant TBM, as probably mono-INH resistant have better prognosis.

- Line 127 and line 174 seem somehow contradictory. If the 5 patients with “unlikely” TBM correspond to “no TBM”, then the authors should exclude them for analysis.

- The authors should explain with more detail why 5 patients were included as DR-TBM without microbiological confirmed resistance. It seems that if clinical course was unfavourable (line 99) some patients were considered as posing DR-TB. This is probably not entirely correct, as many drug-sensitive TBM may have poor outcome not related to the presence of DR-TB.

- Line 127. Please include in the text a brief description of the TBM Consortium case definitions. I think this could help readers to better understand the paper.

- Line 172. 1 patient who was diagnosed by brain biopsy correspond to 0,4% (not 2%)

- Definitions of MDR-TB and XDR-TB have been recently revised; I would consider including definitions used by the authors in the “Methods” section of the paper.

- Have the authors predefined time of follow-up after ending TB treatment?

- It is presented in the “Discussion” section of the text that over 30% of patients died after hospital discharge, but these is not evident in the results section. The same occurs with “80% of LTFU patients subsequently died” (lines 228-230). Please explain in the “Methods” section the definition of “Initial and “Long term” outcome of table-1.

- I have found some errata and acronyms not explained (DST in line 98 –not later in 116-; NRL line 124; IDACIRC line 126, HIV, line 191…)

6. PLOS authors have the option to publish the peer review history of their article (what does this mean?). If published, this will include your full peer review and any attached files.

Reviewer #1: **Yes: **Noyal Mariya Joseph

Reviewer #2: No

---

## [Author Response · Author response to Decision Letter 0]

30 Mar 2022

Dear PLOS One, 

 We very much appreciate you reviewing our manuscript entitled, “Long Term Outcomes of Patients with Tuberculous Meningitis: The Impact of Drug Resistance” and for the constructive feedback. We have included our responses below and made the corresponding changes in the manuscript. Thank you for considering our work, 

Sincerely, 

Russell Kempker on behalf of the study team 

Editorial/Journal comments. 

1. Comment: Please ensure that your manuscript meets PLOS ONE's style requirements, including those for file naming. The PLOS ONE style templates can be found https://journals.plos.org/plosone/s/file?id=wjVg/PLOSOne_formatting_sample_main_body.pdf and https://journals.plos.org/plosone/s/file?id=ba62/PLOSOne_formatting_sample_title_authors_affiliations.pdf.

Response: Thank you for this guidance. We have reformatted the title page, abstract, text, figures and tables and references according to the guidance provided. Please let us know if there are any further revisions needed. 

2. Thank you for stating the following in the Acknowledgments Section of your manuscript: “This work was supported in part by IDSA and HIVMA through a G.E.R.M. award (E.E) as well as support from the NIH including the Fogarty International Center (D43 TW007124) and NIAID (R03 AI139871, K23AI103044, and K23 AI144040).”

“This work was supported in part by IDSA and HIVMA through a G.E.R.M. award (E.E) as well as support from the NIH including the Fogarty International Center (D43 TW007124) and NIAID (R03 AI139871 [RRK], K23AI103044 [RRK], and K23 AI144040[JMC). The funders had no role in the study design, data collection and analysis, decision to publish or preparation of the manuscript.” Correct text to use for our funding statement. 

Response: Thank you for this guidance. We have removed the funding text from the acknowledgement section of the manuscript. We confirm that the text included above in our Funding Statement is correct and should be used as is. 

3. Comment: We note that you have stated that you will provide repository information for your data at acceptance. Should your manuscript be accepted for publication, we will hold it until you provide the relevant accession numbers or DOIs necessary to access your data. If you wish to make changes to your Data Availability statement, please describe these changes in your cover letter and we will update your Data Availability statement to reflect the information you provide.

Response: Thank you for this clarification. We have discussed as a team and will provide the data in excel format after removal of patient identifying information. 

4. Comment: Your ethics statement should only appear in the Methods section of your manuscript. If your ethics statement is written in any section besides the Methods, please move it to the Methods section and delete it from any other section. Please ensure that your ethics statement is included in your manuscript, as the ethics statement entered into the online submission form will not be published alongside your manuscript.

Response: We have removed the ethics statement at the end of the manuscript. It is now only included in the methods section. 

Reviewers' comments:

Reviewer #1

5. Comment: In the results the authors have mentioned drug resistant TB is more common among HIV patients and those treated prior for TB. Can you please perform a statistical test to confirm the validity of these findings and mention the p value?

Response: We have performed a Chi-Square statistical test to compare the prevalence of HIV and prior treatment among drug-susceptible and drug-resistant TB and have included a column of p-values in Table 1 and in the results text. 

Reviewer #2

6. Comment: Baseline characteristics of patients with drug-resistant TB seem to be of worse prognosis than patients with drug-susceptible TB (higher proportion of HIV coinfection, worse neurologic basal status, evidence of extra-CNS disease…) so that may confound for interpreting the outcome. I would suggest the authors to explain in more detail how multivariate analysis was performed, variables included in the multivariate model, and including in table-1 the “p value” on differences of basal characteristics.

Response: Thank you for this comment. We have added p values to table 1 to show whether differences between included characteristics are statistically significant. To describe our model building process we have included the following sentences in the methods, “We calculated survival for drug-resistant and drug susceptible TBM according to the Kaplan–Meier method and used a Cox proportional-hazards model to adjust for between-group differences in baseline characteristics [20, 21].... The main predictor variable was treatment for drug-resistant disease. Other factors with clinical significance in univariate analysis, with biological plausibility, or previously found to be associated with mortality among patients with TBM were considered in the model.

Additionally, we included the additional variables included in the model for mortality in the text and in table 3. Additional variables included HIV and TBM grade (severity of disease) to control for potential confounders. 

7. Comment: It would be of interesting to include also a few data of the outcome of INH-resistant TBM, as probably mono-INH resistant have better prognosis.

Response: Thank you for this comment. There were 4 patients with INH resistance (without rifampin resistance) who died including 3 with HIV. Thus, unfortunately the death rate was similar to the overall death among patients treated for drug-resistant TBM. We therefore did not break up the deaths by category of drug resistance. 

8. Comment: Line 127 and line 174 seem somehow contradictory. If the 5 patients with “unlikely” TBM correspond to “no TBM”, then the authors should exclude them for analysis.

Response: Thank you for this comment. We have clarified this in the manuscript further. The patients who did not meet the scoring criteria for probable TBM had their charts reviewed by two US Infectious Diseases Physicians (thus not involved in patient clinical care) and were included if there were no alternative diagnoses and their presentation was compatible with at least possible TBM. Most of these cases did not meet scoring criteria given lack of information with the main one being the duration of their symptoms (symptoms must be > 5 days to get 4 points in the clinical scoring section). Given the retrospective nature of the study we were not able to get such clinical details from all cases. We felt it was important to include this group given their presentation was compatible with TBM by our chart review and also by the clinical decision making of experience TBM physicians. We have added in the following sentence to the methods to make this point clearer, “Patients who did not meet the case definition of TBM by scoring criteria also had a detailed chart review and those without alternative diagnoses and a clinical presentation compatible with TBM were included”. Given the uncertainty of unconfirmed TBM cases, we have also included analyses limited to probable or definite cases which show similar results to our overall cohort. 

9. Comment: The authors should explain with more detail why 5 patients were included as DR-TBM without microbiological confirmed resistance. It seems that if clinical course was unfavorable (line 99) some patients were considered as posing DR-TB. This is probably not entirely correct, as many drug-sensitive TBM may have poor outcome not related to the presence of DR-TB.

Response: Thank you for this comment and we agree more clarification is needed. In the prior version of the manuscript, we had just included information in the figure 1 legend, mentioning that 5 of the patients were treated for MDR TBM given a known contact with a confirmed MDR TBM case or prior TB treatment failure. We have now added the following sentence in the results to make this clearer, “The five unconfirmed cases were treated for MDR TBM given either prior TB treatment failure or a known contact with confirmed MDR TB.” 

10. Comment: Line 127. Please include in the text a brief description of the TBM Consortium case definitions. I think this could help readers to better understand the paper.

Response: We have added the following sentence to highlight the different scoring domains used to classify patients in TBM categories, “In brief, this case definition uses clinical, CSF, and imaging criteria to provide a score that categorizes patients into possible, probable, or definite TBM [17].”

11. Comment: Line 172. 1 patient who was diagnosed by brain biopsy correspond to 0,4% (not 2%). 

Response: Thank you for this pick up. We have amended to corrected to indicated 0.4%. 

12. Comment: Definitions of MDR-TB and XDR-TB have been recently revised; I would consider including definitions used by the authors in the “Methods” section of the paper.

Response: This is a good point, especially with the updated drug-resistance definitions from WHO released in January 2021. We have used prior definitions and have clarified with this additional information added in the methods, “Patients were categorized as either drug susceptible or drug-resistant TBM with drug-resistance defined as having INH monoresistance (without rifampin), rifampin mono-resistance, MDR, pre-XDR or XDR based on NCTLD and WHO guidance from 2020. MDR was defined as resistance to rifampin and isoniazid; pre-XDR as additional resistance to a fluoroquinolone or injectable second-line agent; XDR as resistance to both a fluoroquinolone and an injectable second-line agent.”

13. Comment: Have the authors predefined time of follow-up after ending TB treatment?

Response: Our length of follow up time was determined by the last date of checking the Georgian national Office of Statistics for the evaluation of death, which was performed on August 29, 2019. We have added this information to the methods under the section entitled, “Data Collection”. 

14. Comment: It is presented in the “Discussion” section of the text that over 30% of patients died after hospital discharge, but these is not evident in the results section. The same occurs with “80% of LTFU patients subsequently died” (lines 228-230). Please explain in the “Methods” section the definition of “Initial and “Long term” outcome of table-1.

Response: To make this clearer we have made some amendments to the text in the methods and also Table 1. We have clarified that end of treatment outcomes were our “initial outcomes” and we have changed long term outcomes in table 1 to long term mortality which is defined and explained in the methods section. We changed the discussion text from >30% of patients died after discharge to >30% died after treatment completion to reflect our results accurately. Additionally, we have included the following lines in the result section to make clear how many people died after treatment and LTFU, “ There were 73 deaths (31%) among our entire TBM cohort including 50 persons who died during treatment and 23 who died after treatment completion. There were 15 persons who were LTFU during treatment after a median of 62 days (IQR 39-138), 12 of whom were determined to have died by death records.”

15. Comment: I have found some errata and acronyms not explained (DST in line 98 –not later in 116-; NRL line 124; IDACIRC line 126, HIV, line 191…)

Response: Thank you for pointing these out. We have gone through and spelled out these acronyms the first time they were introduced.

---

## [Decision Letter · Decision Letter 1]

7 Jun 2022

Long Term Outcomes of Patients with Tuberculous Meningitis: The Impact of Drug Resistance

PONE-D-21-30903R1

Dear Dr. Kempker,

We’re pleased to inform you that your manuscript has been judged scientifically suitable for publication and will be formally accepted for publication once it meets all outstanding technical requirements.

Kind regards,

Pere-Joan Cardona, MD, PhD

Academic Editor

PLOS ONE

Additional Editor Comments (optional):

Reviewers' comments:

Reviewer's Responses to Questions

**Comments to the Author**

1. If the authors have adequately addressed your comments raised in a previous round of review and you feel that this manuscript is now acceptable for publication, you may indicate that here to bypass the “Comments to the Author” section, enter your conflict of interest statement in the “Confidential to Editor” section, and submit your "Accept" recommendation.

Reviewer #2: All comments have been addressed

2. Is the manuscript technically sound, and do the data support the conclusions?

Reviewer #2: (No Response)

3. Has the statistical analysis been performed appropriately and rigorously? 

Reviewer #2: (No Response)

4. Have the authors made all data underlying the findings in their manuscript fully available?

Reviewer #2: (No Response)

5. Is the manuscript presented in an intelligible fashion and written in standard English?

Reviewer #2: (No Response)

6. Review Comments to the Author

Reviewer #2: In my opinion, the authors have correctly addressed our suggestions.

Please revise some errata (i.e, - Line 137 “Drug resistance was defined”)

7. PLOS authors have the option to publish the peer review history of their article (what does this mean?). If published, this will include your full peer review and any attached files.

Reviewer #2: No

---

## [Editor Report · Acceptance letter]

17 Jun 2022

PONE-D-21-30903R1 

Long Term Outcomes of Patients with Tuberculous Meningitis: The Impact of Drug Resistance 

Dear Dr. Kempker:

I'm pleased to inform you that your manuscript has been deemed suitable for publication in PLOS ONE. Congratulations! Your manuscript is now with our production department. 

Kind regards, 

on behalf of

Prof. Pere-Joan Cardona 

Academic Editor

PLOS ONE